# Heat-Mode Excitation in a Proximity Superconductor

**DOI:** 10.3390/nano12091461

**Published:** 2022-04-25

**Authors:** Artem Denisov, Anton Bubis, Stanislau Piatrusha, Nadezhda Titova, Albert Nasibulin, Jonathan Becker, Julian Treu, Daniel Ruhstorfer, Gregor Koblmüller, Evgeny Tikhonov, Vadim Khrapai

**Affiliations:** 1Osipyan Institute of Solid State Physics, Russian Academy of Sciences, 142432 Chernogolovka, Russia; adenisov@princeton.edu (A.D.); anton.bubis@skoltech.ru (A.B.); stanislau.piatrusha@uni-wuerzburg.de (S.P.); tikhonov@issp.ac.ru (E.T.); 2Department of Physics, Princeton University, Princeton, NJ 08544, USA; 3Skolkovo Institute of Science and Technology, Nobel Street 3, 121205 Moscow, Russia; a.nasibulin@skoltech.ru; 4Institute of Physics, Technology, and Informational Systems, Moscow Pedagogical State University, 29 Malaya Pirogovskaya St, 119435 Moscow, Russia; titovana@mail.ru; 5Center for Nanotechnology and Nanomaterials, Walter Schottky Institut and Physik Department, Technische Universität München, Am Coulombwall 4, 85748 Garching, Germany; jonathan.becker@wsi.tum.de (J.B.); julian.treu@wsi.tum.de (J.T.); daniel.ruhstorfer@wsi.tum.de (D.R.); gregor.koblmueller@wsi.tum.de (G.K.); 6Faculty of Physics, National Research University Higher School of Economics, 20 Myasnitskaya Street, 101000 Moscow, Russia

**Keywords:** Andreev reflection, charge–heat separation, shot noise

## Abstract

Mesoscopic superconductivity deals with various quasiparticle excitation modes, only one of them—the charge-mode—being directly accessible for conductance measurements due to the imbalance in populations of quasi-electron and quasihole excitation branches. Other modes carrying heat or even spin, valley etc. currents populate the branches equally and are charge-neutral, which makes them much harder to control. This noticeable gap in the experimental studies of mesoscopic non-equilibrium superconductivity can be filled by going beyond the conventional DC transport measurements and exploiting spontaneous current fluctuations. Here, we perform such an experiment and investigate the transport of heat in an open hybrid device based on a superconductor proximitized InAs nanowire. Using shot noise measurements, we investigate sub-gap Andreev heat guiding along the superconducting interface and fully characterize it in terms of the thermal conductance on the order of $G_{\mathrm{th}}∼e^{2}/h$, tunable by a back gate voltage. Understanding of the heat-mode also uncovers its implicit signatures in the non-local charge transport. Our experiments open a direct pathway to probe generic charge-neutral excitations in superconducting hybrids.

## 1. Introduction

Conversion of a quasiparticle current to the collective motion of a Cooper pair condensate at the interface of a normal metal and superconductor is known as Andreev reflection (AR) [1]. For quasiparticle energies (ε) below the superconducting gap (Δ) (sub-gap quasiparticles, |ε|<Δ), AR is fully responsible for the charge transport across the interface. Conservation of both the number of sub-gap quasiparticles and their excitation energy on the normal side manifests AR as a fundamental example of charge–heat separation in the electronic system. Out of thermal equilibrium, the spatial gradient of a charge-neutral quasiparticle distribution conveys the heat flux [2], which does not penetrate the superconductor and propagates along its boundary with a normal conductor. In this way, ARs mediate the heat conduction via vortex core in s-type superconductors [3] and via neutral modes in graphene [4].

The retro-character of the AR, that is, the propagation of a reflected hole via the time-reversed trajectory of an incident electron, results in a suppression of the heat conduction in the ballistic limit. This obstacle may be overcome by imposing the chirality of the charge carriers in a magnetic field [5,6,7], similar to quantum Hall-based experiments [8], or by going in the regime of specular AR near charge-neutrality point in graphene [9]. In the diffusive limit, counter-intuitively, the heat transport is restored, since moderate disorder scattering effectively increases the number of the conducting modes [10]. In addition, the disorder scattering promotes the relaxation of a charge-mode component into pure heat-mode, by mixing the quasi-electron and quasihole branches via AR. For such a relaxation to occur, a superconducting gap has to vary either in momentum space, as in anisotropic bulk superconductors [11], or in real space [12], as in proximity structures, including in the present experiment. All of this makes the geometry of the Andreev wire [10]—a diffusive normal core proximitized by a wrapped around superconductor—preferable for a sub-gap heat transport experiment.

In this work, we challenge a thermal conductance ($G_{\mathrm{th}}$) measurement in an open three-terminal hybrid device based on a diffusive InAs nanowire (NW) proximitized by a superconducting contact, see the image of one of our samples in Figure 1a. Conceptually similar devices were investigated in the context of Cooper-pair splitters [13,14,15] and, more recently, Majorana physics [16,17,18,19,20,21] with the emphasis on the electrical conductance. The central part of the device represents a few 100 nm long Andreev wires with a partial superconducting wrap, which removes complications arising from the Little–Parks effect [22,23]. In a previous work with the same devices [24], we have demonstrated a charge neutrality of a non-local quasiparticle response, which is direct evidence of the heat-mode excitation regime. Here, we focus on a comparison of local and non-local noise signals, evaluation of thermal conductance and the origin of transport signals in this regime. Our experiments offer a so-far missing experimental tool in the field of non-equilibrium mesoscopic superconductivity [25,26,27,28,29,30] and enable the control of generic charge-neutral excitations in superconducting hybrids.

## 2. Results: Devices and Transport Response

The outline of our experiment is depicted in Figure 1b. A semiconducting InAs nanowire is equipped with a superconducting (S) terminal, made of Al, in the middle and two normal metal (N) terminals, made of Ti/Au bilayer, on the sides. Below, we focus on the data from two devices. In the device NSN-I (NSN-II), the length of the NW underneath the superconductor is 200 nm (300 nm) and the NW segments between the S-terminal and the N-terminals are 350 nm (300 nm) long. In essence, this device layout represents two back-to-back normal metal–NW–superconductor (NS) junctions sharing the same S-terminal. Note the absence of the quantum dots [13,15,19] or tunnel barriers [18] adjacent to the S-terminal, which enables better coupling of the sub-gap states to the normal conducting regions. Throughout the experiment, the S-terminal is grounded, terminal N1 is biased and terminal N2 is floating (or vice versa). Note that grounding of the S-terminal protects the Al from non-equilibrium superconductivity effects [25,31]. The S-terminal serves as a nearly perfect sink for the charge current. At energies below the superconducting gap Δ≈180μeV of Al, the S-terminal cannot absorb quasiparticles [1] and their non-equilibrium population can relax only via diffusion to the N terminals [32], manifesting charge–heat separation. This charge-neutral diffusion flux, which is referred to as the heat flux below, is shown by curly arrows in Figure 1b. One part of the heat flux relaxes via the biased terminal, similar to the usual two-terminal configurations [33,34]. The other part bypasses the S-terminal and relaxes via floating terminal. As we will demonstrate below, this heat flux can be detected by means of shot noise thermometry.

For charge–heat separation via AR, the quality of the InAs/Al interface is important, which we verify in transport measurements. In Figure 1c, we show the local differential conductance $G_{2}$ of the biased junction N2-S in device NSN-II as a function of voltage $V_{2}$ at a temperature T=50mK. Without the magnetic field (*B*), $G_{2}$ exhibits two well-defined maxima at finite $V_{2}$ that diminish with increasing the *B*-field directed perpendicular to the substrate and vanish in B≈20mT simultaneously with the transition of the Al to the normal state. The maxima occur around gap edges $V_{2}=\pm \Delta /\left|e\right|$, where *e* is the elementary charge, and the corresponding increase of $G_{2}$ above the normal state value reaches about 15%. This re-entrant conductance behavior is a property of diffusive NS junctions with a highly transparent interface [35]. Around zero bias in B=0, we generally observed a small reduction of $G_{2}$ by about 10% in all back gate voltage ($V_{g}$) range studied. This guarantees that possible residual reflectivity has a minor effect and ARs dominate over normal interface scattering in our devices.

In Figure 1d, we plot non-local differential resistance $r_{21}=dV_{2}/dI_{1}$, where $V_{2}$ is the voltage on terminal N2, as a function of $V_{1}$. In the normal state $r_{21}$ is featureless and consists of the interface resistance along with a few-Ohm contribution of the Al lead, see the trace in B=50 mT in device NSN-I with $r_{21}\approx 40\;\Omega$. By contrast, in B=0 strong gap-related features develop and $r_{21}$ demonstrates local maximum and minima at the gap edges, see vertical arrows. Note that B=0 behavior is non-universal and depending on $V_{g}$, we have also observed bias asymmetry and sign reversal of the $r_{21}$, see two lower datasets for the device NSN-II. These features are related to the energy dependence of the sub-gap conductance and have a thermoelectric-like origin [36], as will be discussed below. Overall, $r_{21}$ being small compared to the individual resistances of the NS junctions signals that the current transfer length $l_{T}$ is small compared with the width of the S-terminal. We estimate $l_{T}\leq 100\;$nm close to the superconducting coherence length in Al, which sets the lowest possible bound for the $l_{T}$, see Appendix A for the details. $r_{21}$ can be expressed via a non-diagonal element of the conductance matrix [37] as $r_{21}\approx -G_{21}/G_{2}G_{1}$, where $G_{i}\approx G_{ii}$ (i=1,2) are the two-terminal conductances of the NS junctions. $G_{21}∼10^{-2}G_{i}$ is a direct consequence of a charge-neutrality of the non-local response in our devices [24] and proves nearly perfect efficiency of the S contact as charge current sink. The actual sign of the non-local conductance $G_{21}$ can be both negative and positive, as determined by a competition of normal and Andreev transmission processes. Corresponding non-local transmission probabilities are commonly denoted by $T_{21}^{ee}$ and $T_{21}^{he}$, respectively [38]. In the present experiment, at zero bias, we observe a small negative conductance $G_{21}<0$, implying that $\Sigma T_{21}^{ee}>\Sigma T_{21}^{he}$, where sum is performed over the eigenchannels.

## 3. Results: Shot Noise Response

Next, we probe the non-equilibrium electronic populations in both NS junctions using shot noise current fluctuations picked-up in the reflection and transmission configurations sketched in Figure 1b. This measurement is performed using a schematics based on a resonant tank circuit and a home-made low-temperature amplifier. The measurement layout and the calibration procedure are detailed in the Appendix A. Figure 2a demonstrates the noise spectral density measured in terminal N2 as a function of $I_{2}$ at two gate voltages. This configuration, referred to as the reflection configuration, is reminiscent of the usual AR noise in two-terminal devices [33,34], and the measured noise is denoted as $S_{R}$. Experimentally, $S_{R}$ represents the spectral density of the auto-correlation noise of current $I_{2}$ under the bias applied to the terminal N2, while the terminal N1 is maintained DC floating, that is, $S_{R}\equiv S_{22}\;\left(I_{1}=0,I_{2}\right)$. The corresponding experimental layout is depicted in the left sketch of Figure 1b. For comparison, a similar measurement in a reference NS device is shown in Figure 2c. In both devices, the results are qualitatively similar, that is, the $S_{R}$ scales linearly with current and exhibits clear kinks at the gap edges (marked by the arrows). Above the kinks, the diminished slope is the same and it corresponds closely to the universal Fano factor F≡1/3 in a diffusive conductor with normal leads [39,40] $\delta S_{R}/2e\delta I\approx F$, as shown by the dashed lines with a marker “*e*”. This familiar behavior [15] verifies elastic diffusive transport in InAs NWs [41] even at energies well above Δ and ensures quasiparticle relaxation solely by diffusion in contacts. In particular, this observation establishes a solid correspondence between the applied bias voltage and the quasiparticle excitation energy in the present experiment. Namely, a small bias window of [V;V+dV]) corresponds to a creation of electron-like and hole-like quasiparticles with the excess energy of |ε|=|eV|. At sub-gap biases (|V|<Δ/|e|), we observe an important difference being a result of joining an extra N-terminal. While in the NS device the slope expectedly doubles [15,33], see the dotted lines in Figure 2c with the effective charge $e^{*}\approx 2e$ denoted by “*2e*”, in the NSN device, it increases much more weakly and corresponds to $e^{*}\approx 1.6e$ assuming the same *F*. Unlike in SNS junctions [42], a fractional value of $e^{*}$ here is not related to a quasiparticle charge in the superconductor, but reflects an unusual boundary condition for the heat flux underneath the S-terminal, see Ref. [31] and Appendix A for the details. While the doubled $e^{*}$ is a direct consequence of the full reflection of heat flux at the S-terminal [32], its intermediate value means that the missing heat flux in the NSN device is transmitted towards the nearby floating N-terminal. Similar behavior was previously observed in topological insulators [43], however, in the present experiment, the transmitted heat flux is directly measurable, as we show below.

In Figure 2b, we plot the current dependencies of the shot noise measured in transmission configuration, $S_{T}$, that is, the noise at the floating terminal N2. In this configuration, we measure the auto-correlation noise at the DC floating terminal N2 under a finite bias current $I_{1}$, that is, $S_{T}\equiv S_{22}\;\left(I_{1},I_{2}=0\right)$. The corresponding experimental layout is depicted in the right sketch of Figure 1b. Within all investigated $V_{g}$ range, $S_{T}$ steeply increases at small currents followed by pronounced kinks at the gap edges, see the arrows for some of the traces, and keeps increasing much more weakly above the kinks. This behavior of $S_{T}$ is explained as follows. Sub-gap quasiparticles diffusing along the superconductor, and experiencing a few ARs on the way, guide the heat flux via proximitized InAs. Above-gap quasiparticles, however, mostly leave via the S-terminal and their contribution to the transmitted heat signal is minimal. This qualitative picture is proved in the following crosscheck experiment. In the upper part of Figure 2b, the $S_{T}$ signals are compared in B=0 and B=50 mT with the Al in superconducting and normal states, respectively. In the normal state, $S_{T}$ grows weakly at increasing $I_{1}$ without any kinks. Moreover above-gap signal in B=0 roughly reproduces this trend up to a vertical shift at high $I_{1}$. We conclude that this effect is mainly caused by residual normal interface scattering, see also Ref. [24]. Importantly, for sub-gap energies, $S_{T}∼S_{R}$, cf. Figure 2a, whereas non-local charge transport resulted in $|G_{21}|\ll G_{1},G_{2}$. This difference emphasizes the fact that non-equilibrium populations of quasiholes and quasi-electrons are balanced in the proximity region and transmitted noise directly probes the heat-mode excitation. Figure 2b, therefore, demonstrates our main result that at sub-gap energies the proximitized InAs NW supports guiding of heat underneath the S-terminal by virtue of AR processes.

We proceed with a quantitative description of the Andreev heat guiding by solving the diffusion equation for the electronic energy distribution (EED), inspired by a quasiclassical approach [31,32]. In the proximitized region, the boundary conditions take into account ARs for the sub-gap transport and residual normal reflections above the gap. Thermal conductance $G_{\mathrm{th}}$ and interface resistance *r* are the only two parameters that, together with known $G_{1},\;G_{2}$, determine the solution for the EED and the noise temperature $T_{N}$ of the floating NS junction [44]. For convenience, we choose electrical units for the thermal conductance [31] $G_{\mathrm{th}}=e^{2}\nu^{*}D^{*}/L_{S}$, where $\nu^{*}$ is the effective one-dimensional density of states, $D^{*}$ is the diffusion coefficient in the NW region covered by the superconductor and $L_{S}$ is the length of the S-terminal. With this choice, in case of energy-independent $G_{\mathrm{th}}$, one can express the heat flux caused by a small thermal bias δT applied across the proximity region as $\dot{Q}=G_{\mathrm{th}}L_{0}T\delta T$, where $L_{0}=\pi^{2}k_{B}^{2}/3e^{2}$ is the Lorenz number. The details of theoretical modeling can be found in the Appendix A. In Figure 3a, we compare the $T_{N}$ measured in the experiment of Figure 2b (solid lines) with the model fits (dashed lines), where $T_{N}\equiv S_{T}/4k_{B}G_{2}$. Plotted as a function of $V_{1}$ the kinks in $T_{N}$ indeed occur at the gap edges for all $V_{g}$ values, see the vertical arrows. The data are perfectly reproduced, ensuring that our model captures correctly the physics of the Andreev heat guiding effect. The $V_{g}$ dependence of the interface parameter *r* is shown in Figure 3c. We find r∼50Ω, which is consistent with $r_{21}$ in the same device in the normal state, cf. Figure 1d, and almost independent of $V_{g}$. The evolution of $G_{\mathrm{th}}$ at increasing $V_{g}$ is shown by symbols in Figure 3b. The initial growth is followed by saturation at $G_{\mathrm{th}}∼2e^{2}/h$. This is in contrast with a monotonic increase of the electrical conductances $G_{1}$, $G_{2}$ of NS junctions in the same device, see the lines in Figure 3b. We attribute this difference to the impact of superconducting proximity effect that diminishes the density of states stronger at higher carrier densities. Note that while the back-gate sensitivity of $G_{\mathrm{th}}$ is consistent with the behavior of the sub-gap states in the NW region covered on top by the superconductor [45], the microscopic origin of such states and its possible relation, e.g., to the spin-orbit coupling in InAs, goes beyond the scope of the present experiment.

## 4. Results: Non-Equilibrium DC Transport

So far, we have used shot noise measurements to demonstrate sub-gap Andreev heat guiding. In the following, we concentrate on the signatures of this effect in charge transport measurements in the device NSN-II. First, we focus on resistive thermometry based on a weak *T*-dependence of the mesoscopic conductance fluctuations. In Figure 4a we plot the out-of-equilibrium linear response resistance $R_{1}=\partial V_{1}/\partial I_{1}{|}_{I_{1}=0,I_{2}\neq 0}$ of the floating NS junction as a function of $V_{2}$ (see the upper sketch in Figure 4 for the measurement configuration). $R_{1}$ exhibits the same qualitative behavior as the $S_{T}$ before, with much stronger dependence at sub-gap energies, kinks at the gap edges and suppression in *B*-field. Using the equilibrium dependencies $R_{1}\left(T\right)$ for calibration, we converted these data in the effective temperature $T^{*}$ of the floating NS junction and plotted in Figure 4b. The behavior of $T^{*}$ is similar to that of the $T_{N}$ in the device NSN-I, cf. Figure 3a, potentially making this approach an alternative for the detection of transmitted heat fluxes. Note, however, that resistive thermometry slightly underestimates the effect compared to a simultaneously measured $T_{N}$, see Appendix A for the details of the analysis. This may be a result of dephasing that causes averaging of the conductance fluctuations and was not taken into account.

Finally, we investigate non-local *I*-*V* characteristics in the configuration shown in the lower sketch of Figure 4. In Figure 4c, the voltage $V_{2}$ is plotted as a function of $I_{1}$ for three representative values of $V_{g}$. All traces lack full antisymmetry, $V_{2}\left(I_{1}\right)\neq -V_{2}\left(-I_{1}\right)$, moreover, the lower and upper traces exhibit local extrema near the origin, meaning that here the symmetric component dominates the *I*-*V*. This is a signature of the Andreev rectification effect [37], which also caused the asymmetry and sign reversal of $r_{21}$ in Figure 1d. Figure 4d shows the symmetric component of the non-local voltage $V_{2}^{\mathrm{symm}}\equiv \left[V_{2}\left(I_{1}\right)+V_{2}\left(-I_{1}\right)\right]/2$ against $V_{1}$. $V_{2}^{\mathrm{symm}}$ evolves concurrently to the $T^{*}$ and $T_{N}$ with pronounced sub-gap behavior and kinks at $V_{1}\approx \pm \Delta /e$, see vertical arrows. The signal is small, in 1μV range, with both the sign and magnitude demonstrating strong $V_{g}$-dependent fluctuations, in contrast with $T^{*}$ and $T_{N}$. We suggest that the finite $V_{2}^{\mathrm{symm}}$ has a thermoelectric-like origin, analogous to thermopower in Andreev interferometers [36], and results from the thermal gradient that builds up in response to the transmitted heat flux. More rigorously, in the absence of inelastic processes in the present experiment, one should think in terms of a spatial gradient of a non-equilibrium EED [31]. The data in Figure 4d are consistent with $V_{g}$ fluctuations of the Seebeck coefficient in InAs NWs without superconductors [46,47] in the range $\left|S/T\right|∼5\;\mu V/K^{2}$, corresponding fits shown by the dashed lines (see Appendix A for the details). In the present experiment, thermoelectric-like response also comes from the energy dependence of the mesoscopic fluctuations, but it can be additionally affected by the Andreev scattering [37]. Note that the degree of asymmetry of the non-local conductance $G_{21}\propto -\left(dV_{2}/dI_{1}\right)$ caused by this effect (see Figure 1d) is comparable to the data in a Cooper pair splitter [48] and in a tunnel-coupled Majorana device [49,50]. Our thermoelectric interpretation may also be useful in explaining these data.

## 5. Discussion

Our experiment reveals the heat-mode excitation in a proximity superconductor via different experimental signatures. On the one hand, in DC transport, both in the resistive thermometry (Figure 4a,b) and in the non-local Andreev rectification (Figure 4d), the heat-mode non-equilibrium manifests itself through the energy dependence of sub-gap quasiparticle transmission probabilities. These energy dependencies are encoded in the *T*-dependence of the linear-response diagonal elements of the conductance matrix (see the Appendix A for the details) and in the effective Seebeck coefficient. On the other hand, in shot noise, the energy dependence is irrelevant and the data of Figure 3a are perfectly fitted with the energy-independent $G_{\mathrm{th}}$. This difference between the transport and noise approaches is conceptual and lies in the charge-neutral origin of the heat-mode excitation, earlier discussed in Ref. [24]. Below, we briefly analyze the origin of various non-local responses in the present experiment.

Consider for simplicity the case of a single mode NSN device, for which the non-local electrical and thermal conductances are given by $G_{21}=G_{0}T_{21}^{-}$ and $G_{\mathrm{th}}=G_{0}T_{21}^{+}$, where $G_{0}=2e^{2}/h$ and $T_{21}^{\pm }=T_{21}^{he}\pm T_{21}^{ee}$ denote the sum/difference of the non-local Andreev and normal transmission probabilities. The observation of $G_{\mathrm{th}}\gg G_{21}$ implies a predominance of the heat-mode excitation over the charge-mode, that is $T_{21}^{he}\approx T_{21}^{ee}\gg \left|T_{21}^{-}\right|$. In this situation, a weak energy dependence of the transmission probabilities primarily affects the $G_{21}$. Within the first-order expansion $T_{21}^{-}=T_{21}^{-}\left(0\right)+\varepsilon \left(dT_{21}^{-}/d\varepsilon \right)$, therefore, the non-local I−V characteristics acquire symmetric component. Using the formalism of Ref. [38], we obtain for the configuration of the bottom sketch in Figure 4: $V_{2}^{\mathrm{symm}}=-\left|e\right|\left(G_{0}/G_{22}\right)\left(dT_{21}^{-}/d\varepsilon \right){\left(V_{1}\right)}^{2}/2$, or, equivalently, $V_{2}^{\mathrm{symm}}=-\left|e\right|\left(dG_{21}/d\varepsilon \right){\left(V_{1}\right)}^{2}/2G_{22}$. The latter relation is also valid in the multimode case, bridging the effective Seebeck coefficient with the energy dependence of the spectral conductance. Similarly, the energy dependence of the diagonal conductance $G_{22}\left(\varepsilon \right)$ is responsible for the resistive thermometry signal in the configuration of the top sketch in Figure 4. Here, the non-zero term comes from the second derivative $d^{2}G_{22}/d\varepsilon^{2}$, as follows from the derivation given in the Appendix A. Such effects are completely irrelevant for the non-local shot noise measurement in the transmission configuration. Estimated from Figure 4d, the energy dependence of the transmission probabilities can result in ∼1% variation of the $G_{\mathrm{th}}\left(\varepsilon \right)$ within the sub-gap window |ε|<Δ in the device NSN-II. Hence, $G_{\mathrm{th}}\left(\varepsilon \right)\approx const$ and the shot noise in the transmission configuration reads $S_{T}=2\left|eV_{1}\right|G_{\mathrm{th}}$ (at T=0). Note, however, that the energy-independent $G_{\mathrm{th}}$ is puzzling itself and, obviously, contradicts the expected presence of the induced superconducting gap in the proximitized NW region. A microscopic resolution of this puzzle is a difficult theoretical task and goes beyond the scope of the present work.

In summary, we investigated the heat-mode excitation manifesting itself in various non-local responses in NSN proximity devices based on InAs NWs. In DC transport, the non-local signals couple to the heat-mode only indirectly, via a weak and non-universal energy dependence of the spectral conductance. This is in stark contrast with our shot noise approach, which senses the randomness caused by the non-equilibrium EED itself, without the need for any type of spectral resolution [51]. In the same way, the shot noise can also probe excitations of different origin, e.g., spin currents in superconducting spintronics [52], or even valley currents [53], by virtue of spontaneous fluctuations that arise when such currents are fed into the adjacent normal lead [54,55,56,57]. Possible applications are not at all limited to the NW-based material platforms. From this perspective, our experiment establishes a natural background to probe charge-neutral excitations, both above-gap in bulk superconductors and sub-gap in proximity superconductors, including the proposed detection of Majorana zero modes in heat transport [58,59,60,61,62] and, possibly, in measurements of the entanglement entropy [63].

## Figures and Tables

**Figure 1 nanomaterials-12-01461-f001:**
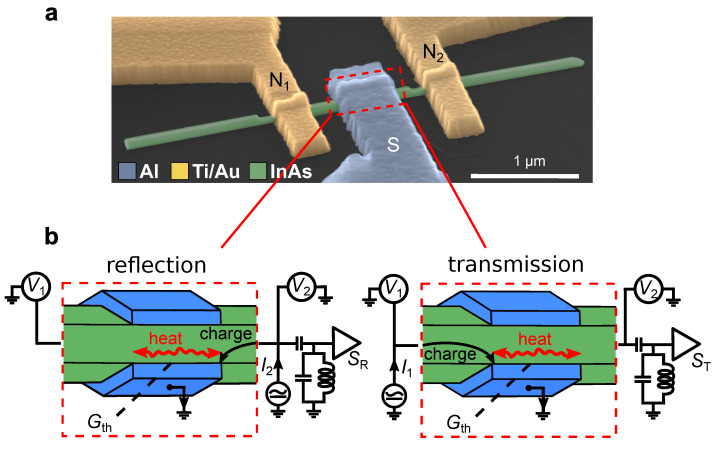
Outline and charge transport data. (**a**) Scanning electron microscope image of the typical device (false color). InAs NW is equipped with two N terminals (Ti/Au) on the sides and one S-terminal (Al) in the middle. (**b**) Separation of charge and heat currents at the InAs/Al interface and two noise measurement configurations. The three-terminal device layout allows studying thermal conductance $G_{\mathrm{th}}$ of the proximitized NW region by measuring shot noise in the transmission configuration. Note that in the present experiment, only terminal N2 is connected to the low temperature amplifier, so that switching between the reflection noise $S_{R}$ and transmission noise $S_{T}$ is achieved by interchanging the biased and floating N-terminals, see the Appendix A for the wiring scheme. (**c**) Local differential conductance of NS junction in device NSN-II measured at T=50mK in different magnetic fields. (**d**) Non-local differential resistance $r_{21}\equiv dV_{2}/dI_{1}$ for two devices plotted at different *B* and $V_{g}$.

**Figure 2 nanomaterials-12-01461-f002:**
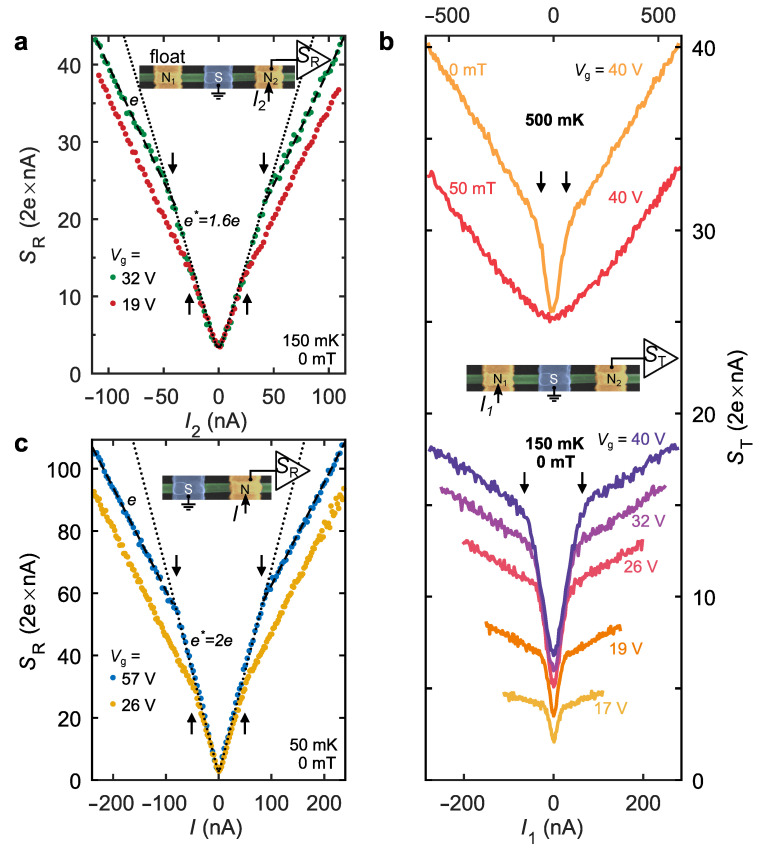
Reflected and transmitted shot noise. (**a**) Reflection noise configuration in device NSN-I. Noise spectral density of the biased NS junction as a function of current at two values of $V_{g}$. Dotted line is the fit with F=0.30 and charge $e^{*}=1.6e$; dashed line slope corresponds to F=0.30 and charge equal to *e*. Green symbols are shifted vertically by $9\times 10^{-28}\;A^{2}/\mathrm{Hz}$ to coincide with red ones at zero bias. (**b**) Transmission noise configuration in device NSN-I. Noise spectral density of the floating NS junction as a function of current at different *B*, *T* and $V_{g}$ (see legend). (**c**) Reflected shot noise in the reference two-terminal NS device as a function of current at two values of $V_{g}$. Dotted line is the fit with $F=\;0.33,\;e^{*}=\;2e$; dashed line slope corresponds to F=0.33 and charge equal to *e*.

**Figure 3 nanomaterials-12-01461-f003:**
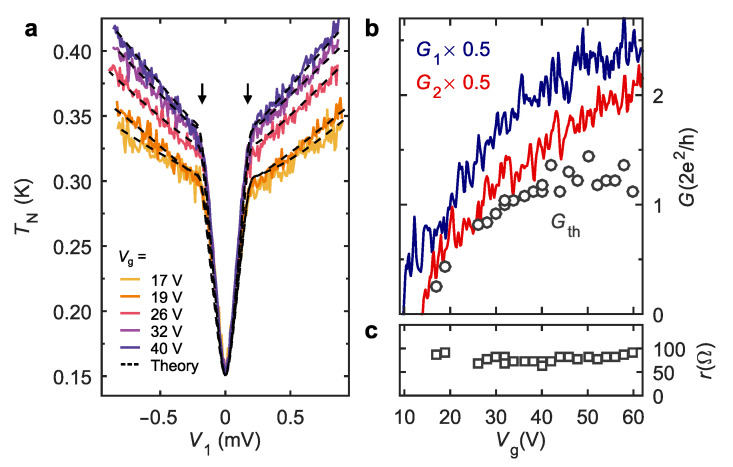
Thermal conductance in the device NSN-I. (**a**) Noise temperature $T_{N}$ measured in the transmission configuration as a function of bias (solid lines, same data as in the lower part of Figure 2b) along with the model fits (dashed lines). (**b**,**c**) (symbols) Sub-gap thermal conductance $G_{\mathrm{th}}$ and interface resistance parameter *r* plotted as a function of $V_{g}$. (lines) Linear response conductances of the left/right ($G_{1}/2$) NS junctions.

**Figure 4 nanomaterials-12-01461-f004:**
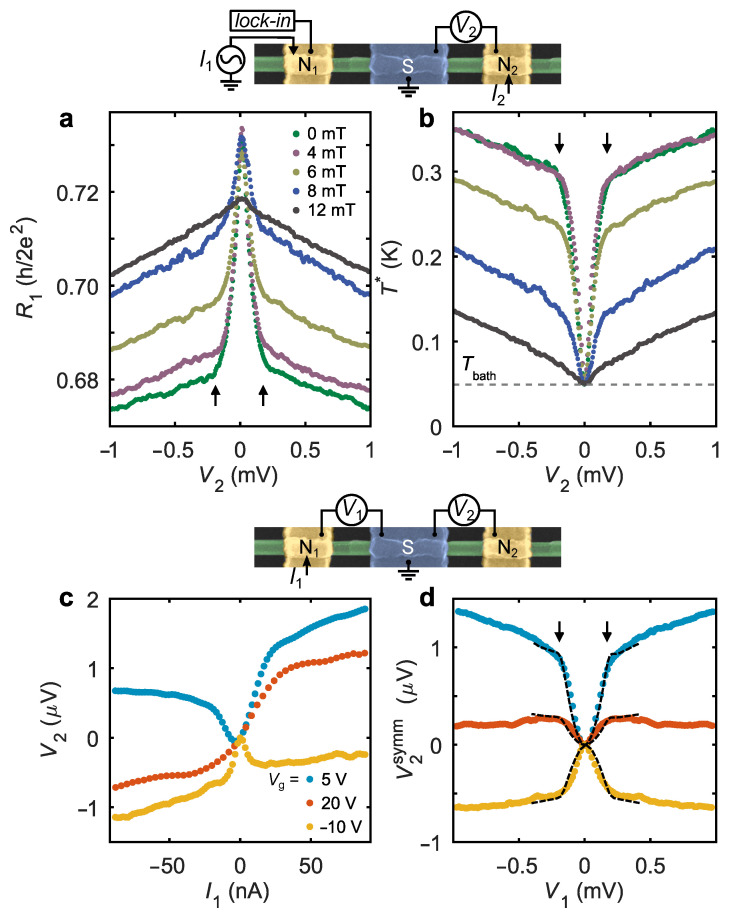
Resistive thermometry and non-local *I*-*V*s in device NSN-II. (**a**) Linear response resistance of the floating NS junction as a function of bias in the neighboring junction. (**b**) The same data converted to the effective temperature $T^{*}$. (**c**) The non-local *I*-*V* characteristics measured at three representative $V_{g}$ values. (**d**) Symmetric component of the non-local *I*-*V*s. The dashed lines are the calculated thermoelectric voltage values for different energy-independent Seebeck coefficients of $S/T=3.0\;\mu V/K^{2},\;0.9\;\mu V/K^{2}$ and $-3.6\;\mu V/K^{2}$ (from top to bottom). Upper sketch: setup for resistive thermometry. Lower sketch: setup for non-local *I*-*V*s.

## Data Availability

The full data for this study can be obtained from the corresponding author upon reasonable request.

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
