# Peer review of "Heat-Mode Excitation in a Proximity Superconductor"

_nanomaterials, 2022, doi:10.3390/nano12091461_

Round 1

Reviewer 1 Report

The manuscript investigates the transport of heat in an open hybrid device based on a superconductor proximitized InAs nanowire.  Using shot noise measurements they investigate sub-gap Andreev heat guiding along the superconducting interface and fully characterize it, even tuned by a back gate voltage.  The experiments open a direct pathway to probe generic charge-neutral excitations in superconducting hybrids.  The results are interesting and suitable for publication in nanomaterials after considering my comments. 
1) The writing of the manuscript needs to be improved. It is really hard to understand to the readers. The meanings of many notations or concepts were not clearly defined, such as the “shot noise measurement”, the sub-gap energy SR,ST and so on.  How these values of SR or ST are connected to the sub-gap energy of their devices based on the circuit design? Why the sub-gap shows back gate dependence?  
2) The manuscript does not give the exact distance, L, between the SC stripe and the N1 or N2. The sub-gap of the proximity on NW must be L dependent.
3) Why the authors use InAs semiconducting NW, not  use a metallic wire instead?
4) In Figure 1, why G2 and G12 are opposite at zero bias in the local and non-local case? 

Author Response

Reply to Referee 1

We thank the Referee for their positive evaluation of the manuscript. We agree that the aspects of the manuscript writing the Referee commented on are worth improving. We revised the manuscript accordingly and hope that the Referee finds this revision reasonable. All the changes in the revised manuscript are highlighted in blue (new inclusions) or red (removed text). Below we reply the remarks of the Referee 1 point by point.

1a. “The writing of the manuscript needs to be improved. It is really hard to understand to the readers. The meanings of many notations or concepts were not clearly defined ...”

Throughout the revised manuscript we have improved the definitions of the physical quantities used. The notions of the sub-gap energy and sub-gap bias are introduced, as well as the definitions of the measured noise spectral densities SR/ST, noise temperature T_N and thermal conductance G_th.

1b. “How these values of SR or ST are connected to the sub-gap energy of their devices based on the circuit design?”

In our manuscript, the term sub-gap energy is used to relate the excess energy of quasiparticles incident from (or escaping to) the N-terminals with respect to the chemical potential of the superconductor. The auto-correlation noises SR and ST directly probe the randomness of the quasiparticle flux in the respective N-terminal.

1c. “Why the sub-gap shows back gate dependence? ”

We believe that the reason for the observed field effect in Gth is that lower facets of the NW remain uncovered by the superconductor. This allows gating of the sub-gap states and could naturally explain the effect. Unfortunately, we could not independently verify this idea in the present experiment.

2. “The manuscript does not give the exact distance, L, between the SC stripe and the N1 or N2. The sub-gap of the proximity on NW must be L dependent.”

All the relevant dimension in both devices are mentioned in the revised manuscript. Unfortunately, the device statistics we have does not allow to verify the dependence of the proximity effect features on the distance between S and N terminals, which equals 350 nm in device NSN-I and 300 nm in device NSN-II.

3. “Why the authors use InAs semiconducting NW, not use a metallic wire instead?”

We use InAs semiconducting NWs instead of metallic wires solely for a technical reason.

Thanks to a much higher resistance in the former case, the tank circuit has higher quality factor that greatly improves the sensitivity of the shot noise measurement. Otherwise, he physics цу discuss is not at all limited to the NW-based material platform, as mentioned in the revised summary paragraph.

4. “In Figure 1, why G2 and G12 are opposite at zero bias in the local and non-local case?”

G2 is a two-terminal conductance that is always positive. The sign of the non-local conductance G21 (note the corrected subscripts in the revised manuscript) can be both positive and negative. As we detail in the revised manuscript, the sign is determined by a competition of normal (T21^ee) and cross-Andreev (T21^he) non-local transmission probabilities. G21<0 corresponds to the case T21^ee> T21^he.

Reviewer 2 Report

The paper is well written and very easy to follow. All the experiments are clearly presented. The results are original and interesting. I think the manuscript can be definitely published in its present form. 

Author Response

Reply to Referee 2

We are grateful to the Referee 2 for their positive evaluation of our work.

Reviewer 3 Report

The manuscript entitled “Heat-mode excitation in a proximity superconductor” describes experimental method in the field of non-equilibrium mesoscopic superconductivity and the control of generic charge-neutral excitations in superconducting hybrids. The Andreev reflection (AR) has the limitation of the heat conduction due to the propagation of a reflected hole via the time reversed trajectory of an incident electron. The manuscript is well written and characterized the method of thermal conductance measurement in an open three-terminal hybrid device. Therefore, I do recommend current manuscript to be published in nanomaterials with some point of revisions.

Comment 1

There is no conclusion. The authors should carefully summarize and list the key conclusions from this research work which should be supported by the most representative research results data.

Comment 2

In this article, there is no mention of the formula for calculating the value. The authors should include the formula and explain it in more detail.

Author Response

Reply to Referee 3

We thank the Referee 3 for their positive evaluation of our manuscript. The two comments raised by the Referee are very relevant and are thoroughly addressed in the revised manuscript. All the changes in the revised manuscript are highlighted in blue (new inclusions) or red (removed text). We detail our point by point replies below.

1. “There is no conclusion. The authors should carefully summarize and list the key conclusions from this research work which should be supported by the most representative research results data.”

We thank the Referee for this very important remark. In the revised manuscript we introduced new section “Discussion”, where our key findings are summarized and explained in more detail. In particular, we discuss why the energy dependence of transmission probabilities is relevant for DC responses and why this is not important for the non-local shot noise measurement in the regime of the heat-mode excitation. We believe that the manuscript benefits from these inclusions.

2. “In this article, there is no mention of the formula for calculating the value. The authors should include the formula and explain it in more detail.”

As far as we understand, in this comment the Referee requests a more focused discussion of the heat conductance (Gth). We thank the Referee for pointing out this important caveat. In response, we have modified the manuscript in two respects. First, we have expressed Gth via a renormalized diffusion coefficient and density of states in the proximitized NW and connected the Gth with a hypothetical thermal transport experiment. Second, we have discussed a relation between the Gth and non-local transmission probabilities, explained why the energy dependence is irrelevant for the Gth and presented a formula for the non-local shot noise ST and Gth in the regime of heat-mode excitation. We believe that altogether these inclusions solidify our argumentation.